# Contemporary trends of witchcraft accusations and resulting violence against children: A scoping review and bibliometric analysis protocol

Cara Spence[1]*, Edward Salifu Mahama[2], Kimberly Jarvis[3], Mary Zettl[1], Vida Nyagre Yakong[4], Mary Ani-Amponsah[4], Helen Vallianatos[5], Samuel Adjorlolo[4], Courage Kosi Setsoafia[2], Geoffrey Maina[6], Solina Richter[6], Pammla Petrucka[6]

1 Department of Medicine, University of Saskatchewan, Saskatchewan, Canada, 2 Department of Languages and Information Studies, University of Development Studies, Tamale, Ghana, 3 Faculty of Nursing, Memorial University of Newfoundland, St. John's, Newfoundland and Labrador, Canada, 4 School of Nursing and Midwifery, University of Ghana, Accra, Ghana, 5 Department of Anthropology, Edmonton, University of Alberta, Alberta, Canada, 6 College of Nursing, University of Saskatchewan, Saskatchewan, Canada

☙ These authors contributed equally to this work.
* cara.spence@usask.ca

## Abstract

### Objective

This review seeks to understand the global trends of contemporary witchcraft accusations and related harms against children and adolescents (0–18 years of age).

## Introduction

Witchcraft-related violence against children and adolescents (children) reflects an alarming and understudied phenomenon of socio-culturally legitimated harm around the globe, particularly in sub-Saharan Africa. 'Witchcraft' explains the unexplainable, such as strokes of luck and/or misfortune. Witchcraft accusations are linked to illness, sudden death, financial misfortune, miscarriages, financial windfall, disability, birth abnormalities, or rare conditions. Religious entities also levy witchcraft accusations, referring to black magic, evil, works or malicious spirits, to profit off families while harming the accused. These accusations result in marginalization, alienation, slandered reputation, communal expulsion, and violence, causing disfiguration, disability, and death. Children are especially vulnerable to witchcraft-related violence, including human trafficking, and ceremonial and cultural sacrifice.

## Inclusion criteria

This scoping review will examine witchcraft accusations and related harms against children and adolescents (0–18 years of age) globally from 1946 to 2024.

which permits unrestricted use, distribution, and reproduction in any medium, provided the original author and source are credited.

**Data availability statement:** No datasets were generated or analyzed during the current study. All relevant data from this study will be made available upon study completion.

**Funding:** University of Saskatchewan, Partnership Development Fund and Social Sciences and Humanities Research Council (SSHRC).

**Competing interests:** The authors have declared that no competing interests exist.

## Exclusion criteria

This scoping review excludes articles that do not report specifics of the accusation, situation, result, age of the accused, or country of origin.

## Methods

This scoping review will follow the Joanna Briggs Institute's Preferred Reporting Items for Systematic Reviews and Meta-Analyses Extension for Scoping Reviews (PRISMA-ScR) Statement. Articles published from January 1, 1946 to December 31, 2024 will be collected across academic, grey literature and web-based databases. A systematic search strategy will be applied in each database, and all search results recorded. A bibliometric analysis will also be undertaken to systematically and rigorously review the extant literature.

Findings of this review will identify areas of collaboration and gaps for further exploration. The literature analysis can raise awareness and inform resource development across health care, education, social work, government, and community sectors to better support victims of witchcraft-related harms.

---

## Introduction

Globally, approximately one billion children and adolescents (herein referred to as children) annually experience some form of violence [1]. The maltreatment of children is a significant problem, occurring most often in the home or within the community [2]. In sub-Saharan Africa (SSA), a considerable amount of mistreatment against children is triggered by witchcraft beliefs and accusations [3]. While witchcraft accusations and related violence have historically focused primarily on women, the trend has shifted to children over the last decade [4,5].

Witchcraft is the belief in supernatural causes for both good luck and misfortune [6]. The belief in unseen forces existed before the age of modernity and colonialist positivist science. As a generalized term used in literature, media, and official reports, witchcraft accusations refer to an individual being labelled as a 'witch' or using 'witchcraft' to cause a negative outcome or occurrence that is not understood or 'rationally' explained through the prevailing belief system or worldview. Related to other terms, such as 'juju', 'fetish', 'black magic', and 'spirits', witchcraft accusations can be linked with sudden deaths, illnesses, financial hardship, failing businesses, infertility, and even displacement due to conflict [7]. The accused are believed to have used "witchcraft" or conducted "a personal act of one individual drawing upon supernatural powers to harm another" [8], p. 344. Accusations are often levied by family, friends, or community members, and those accused are subjected to socially legitimized detainment, community expulsion, violence, or death. Accusations stigmatize victims for life, subjecting them to projections of fear, hatred, evil, avoidance, and recurrent acts of violence or vengeance.

The stigma and impacts of witchcraft accusations are rooted in cultural beliefs and institutional contexts wherein such accusations arise. The accused are subjected to the projection of envy, jealousy, greed, hatred, rivalry, or vengeance, typically over strained or dysfunctional social, political, socio-cultural, or economic conditions. Described as the "dark side of kinship" [9], p. 4, these accusations are often made within close familial or communal relationships when a sudden misfortune, loss, or disease outbreak occurs. Moreover, children may be used as penance for a transgression of another family member, for rituals as a sacrifice, trafficked by charismatic spiritual leaders or others, disallowed from school attendance, or forced into servitude or street begging [4,7].

Children who demonstrate misbehaviour, irritation, undiagnosed mental health episodes, disability, epilepsy, albinism, and even breech or multiple births can be accused of witchcraft or being a witch. Infanticide, sexual assault, and belief in magical powers of the body parts of those with rare conditions also fall within the scope of witchcraft-related violence, expressed as ritualistic rather than punitive [10]. Accusations also result in removal from school, particularly for girls, hindering opportunities for future advancement, often leading to forced servitude, street-engaged living, or banishment [4,7,10].

Cases of witchcraft accusations and related violence against children have been documented by international organizations, community organizations, regional governing bodies [11], academics, and journalists. However, a comprehensive picture of the scope and magnitude of such culturally legitimized practices has not been undertaken despite an estimated "tens of thousands" of children harmed by practices related to witchcraft and ritual attacks [10]. Four major documents published by UNICEF [12], the Pan African Parliament Report [11], and two from the African Child Policy Forum in 2016 and 2022 [2,10], are the most comprehensive and recent publications to date examining the trend of witchcraft-related accusations. There is a lack of literature on longstanding and alarming trends of culturally legitimated violence against children, which is facilitated through deeply rooted belief systems and worldviews, and exacerbated through religious beliefs of Christianity and Islam as practiced on the African continent. According to UNICEF, roughly 13,500 children are accused of witchcraft in the Democratic Republic of Congo every year, resulting in them being shunned, expelled from communities, and precariously vulnerable to violence and harm [13].

A rapid review of studies published on the topic since January 1, 2000 garnered limited results overall. A search of the Cochrane Database of Systematic Reviews and JBI Evidence Synthesis, Academic Search Complete, Global Health, MEDLINE, Public Health, PubMed, Social Services Abstracts, and Gale Academic One File was conducted using terms witch, child, scoping review and/or systematic review. No current protocol or systematic or scoping reviews on the topic of witchcraft beliefs, witchcraft-related violence, or witchcraft-related violence against children were found. Two related systematic or scoping reviews connected to witchcraft resulted from all searches: a systematic review on contemporary views on dementia as witchcraft in sub-Saharan Africa [14] and a scoping review of perceptions, attitudes, and cultural understandings of mental health in Nigeria [15]. Neither of these articles fits the criteria of this review. The scarcity of studies may reflect a lack of awareness, prioritization, and/or reservation to consider this culturally significant taboo and value-laden topic. An examination of this phenomenon of violence against children is critical.

The objective of this scoping review is to gather extant evidence regarding witchcraft accusations and related violence or harm against children and adolescents (0–18 years of age). Rather than confirm whether or not there is a belief, we seek to establish a comprehensive corpus on the topic of witchcraft accusations and related harms against children and adolescents and compile (1) accusation contexts and details of the accused, (2) identities of the accused and accuser(s), (3) resulting harm inflicted on the accused, and (4) resulting impacts on the accused.

## Review questions

The primary review question is: *To what extent does harm or violence against children and adolescents (ages 0–18) result from witchcraft-related accusations in the global context?*

Secondary review questions to inform the primary review question are:

1) What is the prevalence of perceptions, beliefs, and lived experiences of "witchcraft" accusations and actions against children and adolescents (ages 0–18)?

 a) What was the specific accusation made against the victim(s)?

 b) Under what circumstances are accusations made?

 c) Who were the accusers, and what were the relationships with the accused?

 d) What were the resulting outcomes of the accusation? (e.g., type(s) of harm)

 e) What are the lasting impacts on the accused (related to the accusation, and the resulting harm?

2) How do accusations of "witchcraft" impact children and adolescents, specifically in terms of social welfare, well-being, safety, gender-based violence, and socio-economic vulnerability?

3) What historical legacies, customary laws, tribal practices, public discourse, and social and religious beliefs support allegations and facilitate witchcraft-related harms within the country or region?

4) What tools, resources, and approaches have been implemented to address the harmful impacts and outcomes of witchcraft practices and allegations against children and adolescents globally?

Topic: Witchcraft-related violence against children and adolescents

## Eligibility criteria

Eligibility criteria for this scoping review was detailed using the JBI People, Concept, Context (PCC) framework for Scoping Reviews [16] (See Supporting Material S2 File)

### Participants

Children and adolescents (0–18 years of age) globally, with particular interest in the sub-Saharan Africa region.

### Concept

Academic and grey literature reporting on the phenomenon of witchcraft-related accusations and violence or harms against children and adolescents associated with these beliefs and practices. Witchcraft is understood in the broad sense to include accusations or actions inciting the unseen or the irrational to induce an outcome, typically negative in result. Sources involving children (anyone under 18 years of age) and adolescents (those from ages 10–18 years of age) who were subject to allegations or consequences of allegations will be included. Harm or violence includes persecutory violence, such as physical violence; social and psychological harm, including stigma, isolation, and psychological trauma; spiritual or supernatural harm which holds power within cultural contexts, influencing fear, social control, and justifies violence; and institutional and systemic violence, including laws that criminalize witchcraft.

Inclusion criteria: children and adolescents aged 0–18 years, any country, any language, synonymous terms including 'juju', 'fetish', 'black magic', 'spirits'; reports, grey material or literature published January 1, 1946 to December 31, 2024.

Exclusion criteria: no report of the age of the accused; missing details on the context and outcome of the accusations; no mention of allegations of witchcraft or related terms.

 

## Context

Globally, any country or region; specific attention to SSA; published in any language.

The scoping review seeks to understand the phenomenon of documented witchcraft accusations and the related violence against children emanating from the beliefs underpinning them. This review aims to 1) shed light on the topic of witchcraft-related violence against children and adolescents; 2) clarify the scope and context of the issue globally, specifically within SSA; 3) contribute to the literature on the topic to enhance the awareness and knowledge of the issue among frontline health care providers, educators, resettlement and social workers, community-based and governmental decision-makers who may encounter witchcraft beliefs and identify children and adolescents at-risk or exposed to witchcraft-related violence in the home or within their communities.

The bibliometric analysis will seek to identify emergent research hot spots (i.e., topics, geographic presence), gaps, evidence-based practices, research pathways (i.e., policy, cultural indicators), distribution and productivity of researcher/collaboratives on the topic.

## Types of sources

Witchcraft is a societal issue framed by cultural worldviews. This scoping review will consider cross-cultural, descriptive observational study designs, systematic reviews, case series, prospective and retrospective cohort studies, reports, case studies, individual case reports, and descriptive cross-sectional studies extracted from relevant academic literature databases and considered for inclusion. Study designs include phenomenology, grounded theory, ethnography, qualitative description, action research, social work theory, and feminist research. Text, opinion papers, official reports, thesis, and 'grey literature' will be extracted from a worldwide Google web search and grey literature databases to be considered for inclusion.

## Materials and methods

This review will use the Joanna Briggs Institute methodology for scoping reviews [16]. The protocol is registered with the Open Science Framework (https://doi.org/10.17605/OSF.IO/P4ATC).

### Search strategy

Studies will be reviewed for inclusion criteria. When a study is published in another language, Google Translate will be used to translate the abstract to assess its validity. If the article meets the criteria, we will further translate it with Google Translate and extract relevant data, provided the translation is sufficient to understand the study in its entirety. (Additional search strategy details found in Supporting Information S3 File.)

The databases to be searched include:

Academic Search Complete; Global Health; MEDLINE (via OVID); Public Health; PubMed; Social Services Abstracts; JSTOR; OARE; Project Muse; Ingenta Connect; Scopus; eHRAF World Cultures; Gale Academic One File; Embase, CINAHL Plus, APA PsychInfo (via EBSCOhost), Anthropology, ERIC, SocIndex, Women's Studies International, International Bibliography of the social sciences, Latin American and Caribbean Health Sciences Literature, African Journals Online, African Index Medicus (AIM),

Grey Literature will be searched across:

ProQuest Dissertations and Theses, OATD: Open Access Theses and Dissertations, Bielefeld Academic Search Engine, OAIsterm UGSpace; Google Scholar; GreyGuide; OpenGrey; Grey Source Index.

Titles and abstracts will be screened by two or more independent reviewers for assessment against the inclusion criteria for the review. A third reviewer will review the conflicts. Potentially relevant sources will be retrieved in full and imported into Covidence, a web-based tool used to streamline the literature review process by assisting with reference screening,

data extraction, and project tracking. PDFs of relevant sources will be collated and kept in a shared file accessible to reviewers, and managed by the corresponding author.

Two or more independent reviewers will thoroughly assess the full text of selected citations against the inclusion criteria. Reasons for excluding sources of evidence will be recorded and reported in the scoping review. Any disagreements between the reviewers at each stage of the selection process will be resolved through discussion and consensus or with an additional reviewer. The results of the search and the study inclusion process will be reported in full in the final scoping review and presented in a Preferred Reporting Items for Systematic Reviews and Meta-analyses Extension for Scoping Review (PRISMA-ScR) flow diagram [17]. *This scoping review protocol describes a systematic synthesis of existing literature and does not involve the collection of primary data from human or animal subjects; therefore, ethical review from an Institutional Review Board (IRB) is not required. The review will adhere to established guidelines for systematic reviews, specifically the PRISMA-ScR checklist and guidelines, ensuring rigorous methodology and transparent reporting of findings.*

### Data extraction

Data will be extracted from papers included in the scoping review by two or more independent reviewers using a data extraction tool developed and tested by the reviewers. The data extracted will include specific details about the age of the accused, country of location, ethnicity, cultural or religious background, language, literacy level, urbanicity (if reported), context which prompted an accusation, the accuser and their relationship to the accused, the resulting harm or action taken against the accused, and outcome on the accused. We will also remain open to any study methods and key findings relevant to the review question and sub-questions.

The data extraction tool will be derived from JBI's data extraction tool template [18] and modified according to this scoping review's research questions and eligibility criteria. The modified tool will be reviewed, pilot tested with a sampling of 10 articles, refined as necessary, and validated before full data extraction begins. The extraction may require additional modifications for a full-text review to be conducted. In this case, the needed changes will be discussed and mutually agreed upon before modifying the extraction tool. A modified draft extraction form is provided in the Supporting Information (see S4 File). It will also incorporate the PRISMA checklist extensions for Scoping Reviews, and reviews reporting harms [19,20]. Modifications will be further detailed in the scoping review results. Any disagreements that arise between the reviewers will be resolved through discussion, or with an additional reviewer/s. If appropriate, authors of papers will be contacted to request missing or additional data, where required.

### Data analysis and presentation

A quantitative count and categorization of evidence types will be presented by bibliographic mapping of relevant literature. The mapping of bibliographic data will be conducted by a librarian expert and two members of the research team using appropriate tools, such as Bibliometrix and HistCite. Data points will be compiled, completing a central objective of this review. The data points will highlight the demographic factors among children and adolescents who are most at risk or experience the highest rates of violence linked to witchcraft accusations. The literature mapping will also illustrate situations that prompt witchcraft accusations, the type(s) of violence, harm, or action that results, the relationship of the accusers to the accused, and the outcome(s) for the accused.

The findings will be presented as a descriptive (or narrative) scoping review and bibliometric map to inform a community-based project investigating witchcraft-related violence in Ghana. The findings will also inform resource development for frontline workers regarding witchcraft beliefs and the risks children and adolescents may face. This work will further potentiate cultural adeptness of primary care and frontline workers engaging with newcomers or those from cultures where witchcraft beliefs are prevalent.

## Supporting information

**S1 File. Preferred reporting items for systematic reviews and meta-analyses extension for scoping reviews (PRISMA-ScR) checklist.**
(DOCX)

**S2 File. Scoping review PCC.**
(DOCX)

**S3 File. Literature search strategy.**
(DOCX)

**S4 File. Data extraction tool (Modified from JBI Template).**
(DOCX)

**S5 File. PRISMA-P-checklist - PONE-D-25-33597.**
(PDF)

## Author contributions

**Conceptualization:** Cara Spence, Edward Salifu Mahama, Kimberly Jarvis, Vida Nyagre Yakong, Mary Ani-Amponsah, Helen Vallianatos, Courage Kosi Setsoafia, Solina Richter, Pammla Petrucka.

**Funding acquisition:** Cara Spence, Kimberly Jarvis, Solina Richter.

**Methodology:** Cara Spence, Kimberly Jarvis, Mary Zettl, Solina Richter, Pammla Petrucka.

**Supervision:** Cara Spence, Solina Richter.

**Writing – original draft:** Cara Spence.

**Writing – review & editing:** Edward Salifu Mahama, Kimberly Jarvis, Mary Zettl, Vida Nyagre Yakong, Mary Ani-Amponsah, Helen Vallianatos, Samuel Adjorlolo, Geoffrey Maina, Solina Richter, Pammla Petrucka.

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
