## [Decision Letter · Decision Letter 0]

11 Jan 2026

Contemporary trends of witchcraft accusations and resulting violence against children: A Scoping Review and Bibliometric Analysis Protocol

PONE-D-25-33597

Dear Dr. Spence,

We’re pleased to inform you that your manuscript has been judged scientifically suitable for publication and will be formally accepted for publication once it meets all outstanding technical requirements.

As a Peer-Reviewed Funded Protocol, your submission is eligible for expedited review by in-house editors. Based on our evaluation, we are satisfied that your manuscript meets our publication criteria for Study Protocols, and is therefore considered to be suitable for publication subject to final journal requirements.

Kind regards,

Steve Zimmerman, PhD

Senior Editor, PLOS One

Reviewers' comments:

Reviewer's Responses to Questions

**Comments to the Author**

1. Does the manuscript provide a valid rationale for the proposed study, with clearly identified and justified research questions?

Reviewer #1: Yes

Reviewer #2: Yes

2. Is the protocol technically sound and planned in a manner that will lead to a meaningful outcome and allow testing the stated hypotheses?

Reviewer #1: Yes

Reviewer #2: Yes

3. Is the methodology feasible and described in sufficient detail to allow the work to be replicable?

Reviewer #1: Yes

Reviewer #2: Yes

4. Have the authors described where all data underlying the findings will be made available when the study is complete?

Reviewer #1: Yes

Reviewer #2: Yes

5. Is the manuscript presented in an intelligible fashion and written in standard English?

Reviewer #1: Yes

Reviewer #2: Yes

You may also provide optional suggestions and comments to authors that they might find helpful in planning their study.

Reviewer #1: This is an interesting topic of publich health concern, particularly in sub-Saharan African Region. The problem is actual and deeply embedded in african traditions, some of which are harmful and need to be exposed. It is a sensitive topic with hard to find literature, being tabou for discussion in many settings and resistant to behavioral change. It will be interesting to read the results of the study.

Line 38: ...against children and adolescents (children). Why children again in brackets? Adolescents will be addressed as children in this manuscript, but this sentence rather looks like a typo and can be unclear for the reader. A better (suggested) way may have been to write …against children (0-18…like in the objectives or inclusion criteria. Or adolescents under the age of 19, considering that 18 years is already adult age.

Reviewer #2: The research topic is truly unique, the methodological structure is precise, and the sources relied upon and analyzed are organized and consistent with the research objectives.

**Do you want your identity to be public for this peer review?** For information about this choice, including consent withdrawal, please see our Privacy Policy

Reviewer #1: **Yes:** Maria Afadapa

Reviewer #2: No

---

## [Editor Report · Acceptance letter]

PONE-D-25-33597

PLOS One

Dear Dr. Spence,

I'm pleased to inform you that your manuscript has been deemed suitable for publication in PLOS One. Congratulations! Your manuscript is now being handed over to our production team.

Kind regards,

on behalf of

Dr Steve Zimmerman

Staff Editor

PLOS One